# Bandit Learning with Joint Effect of Incentivized Sampling, Delayed Sampling Feedback, and Self-Reinforcing User Preferences

**Tianchen Zhou[1], Jia Liu[1], Chaosheng Dong[2], Yi Sun[2]**
[1]Department of Electrical and Computer Engineering, The Ohio State University
[2]Amazon, Seattle, Washington, USA
zhou.2220@osu.edu, liu@ece.osu.edu,{chaosd, yisun}@amazon.com

## Abstract

In this paper, we consider a new multi-armed bandit (MAB) framework motivated by three common complications in online recommender systems in practice: (i) the platform (learning agent) cannot sample an intended product directly and has to incentivize customers to select this product (e.g., promotions and coupons); (ii) customer feedbacks are often received later than their selection times; and (iii) customer preferences among products are influenced and reinforced by historical feedbacks. From the platform's perspective, the goal of the MAB framework is to maximize total reward without incurring excessive incentive costs. A major challenge of this MAB framework is that the loss of information caused by feedback delay complicates both user preference evolution and arm incentivizing decisions, both of which are already highly non-trivial even by themselves. Toward this end, we first propose a policy called "UCB-Filtering-with-Delayed-Feedback" (UCB-FDF) policy for this new MAB framework. In our analysis, we consider delayed feedbacks that can have either arm-independent or arm-dependent distributions. In both cases, we allow unbounded support for the random delays, i.e., the random delay can be infinite. We show that the delay impacts in both cases can still be upper bounded by an additive penalty on both the regret and total incentive costs. This further implies that logarithmic regret and incentive cost growth rates are achievable under this new MAB framework. Experimental results corroborate our theoretical analysis on both regret and incentive costs.

## 1 Introduction

In recent years, the multi-armed bandit (MAB) framework has received a significant amount of interest in the learning research community. This is partly due to the fact that, in many online e-commerce recommender systems (e.g., Amazon and Walmart), the problem of online learning of the optimal products while making profits at the same time can be well formulated by an MAB problem. However, although many MAB algorithms have been proposed in this area, it is worth noting that most of the existing MAB models in the literature have not considered the joint effect of *three* common phenomena in e-commerce recommender systems: (i) In many e-commerce recommender systems, the platform (the learning agent) cannot sample an intended product (an intended arm) directly and has to incentivize customers (e.g., through promotions and coupons) to sample the product and receive the sampling feedback from the customers indirectly (e.g., ratings and reviews); (ii) Customer feedbacks are often received much later than their purchasing times (e.g., a review may or may not be submitted by a customer even months later after purchasing a product); and (iii) Customer preferences among products are influenced and reinforced by historical feedbacks, which may even lead to various viral effects over some products (the more good reviews one product has received, the more likely that the next arriving customer will prefer this product). The lack of a fundamental understanding and joint studies of these three important factors in MAB policy designs motivates us to fill this gap in this paper.

Toward this end, we propose a new MAB framework that *jointly* considers i) incentivized sampling, ii) delayed sampling feedback, and iii) self-reinforcing user preferences in online recommender sys-

tems. However, we note that the MAB policy design for the proposed new MAB framework is highly non-trivial due to the complex couplings between the aforementioned three factors. First, similar to conventional MAB problems, there exists a dilemma between sufficient *exploration* through sampling to learn an optimal arm (i.e., an optimal product), which may incur numerous pullings of sub-optimal arms, and the greedy *exploitation* to play the arm that has performed well thus far to earn profits. Second, there is another dilemma to the learning agent between offering sufficiently attractive incentives to mitigate biases (due to lack of initial data and self-reinforcing user preferences) and avoid spending unnecessarily high incentives that hurt the learning agent's profits. Last but not least, the delayed sampling feedbacks may render the estimation of arms' quality during the MAB process highly inaccurate, introducing yet another layer of uncertainty to the MAB online learning problem, which is already plagued by complications from incentivized sampling and self-reinforcing user preferences. As in most MAB problems, we adopt "regret" as our performance metric in this paper, which is defined as the cumulative reward gap between the proposed policy and an optimal policy design in hindsight. Under the regret setting, the complications due to these three key factors naturally prompt the following fundamental questions:

(1) How should the agent design an incentivizing strategy to strike a good balance between exploration and exploitation to achieve sublinear (hopefully logarithmic) regrets?

(2) To avoid offering exceedingly high incentives, how should the agent incentivize in order to attract a user crowd that prefer an optimal arm, so that the users' self-reinforcing preference could automatically gravitate toward this optimal arm without further incentives?

(3) Under various delayed feedback situations in the new MAB framework (e.g., unbounded random delays, heavy-tailed delay distributions, and arm-dependent delays), could we still achieve low regrets with low incentive costs?

In this paper, we answer the above fundamental questions affirmatively by proposing a new "Delayed-UCB-Filtering" policy for the MAB framework that jointly considers incentivizing sampling, delayed sampling feedback, and self-reinforcing user preferences. We show that our proposed policy achieves $O(\log T)$ regret with $O(\log T)$ incentive payments. The success of our policy design hinges upon two key insights: (i) the self-reinforcing user preference effect is actually a "blessing in disguise" and can be leveraged to establish an important "dominance" condition (more on this later) that further implies $O(\log T)$ regret and incentive costs; and (ii) the impacts of delayed feedback on regret and incentive costs can be upper bounded under appropriate statistical settings to preserve the "dominance" condition. Our key contributions and main results are summarized as follows:

• We propose a new MAB model that jointly considers incentivized arm sampling, delayed sampling feedback, and self-reinforcing user preferences, all of which are important features of online recommender systems. To develop efficient and low-cost incentivized policy for this new MAB model, we propose a three-phase "UCB-Filtering-with-Delayed-Feedback" (UCB-FDF) policy, which contains an incentivized exploration phase, an incentivized exploitation phase, and a self-sustaining phase. In our UCB-FDF policy, the first two phases judiciously integrate delayed feedback information, while in third phase, the system solely relies on self-reinforcing user preferences to converge to the pulling of the optimal arm.

• We first show a fundamental fact that, under our UCB-FDF policy, delayed sampling feedback only has an *additive* penalty on the regret and incentive cost performances, and that this additive penalty grows logarithmically with respect to time. Specifically, we first investigate the delayed feedback impact under the assumption that the feedback delay is an i.i.d. random variable across samplings with a finite expectation. We show that the UCB-FDF policy achieves logarithmic growth rates of regret and incentive costs under this setting. Then, we relax the i.i.d. feedback delay assumption to allow the feedback delay distribution to be arm-dependent. Under this setting, we also show that similar logarithmic growth rates of regret and incentive can still be achieved.

• We conduct extensive experiments on Amazon Review Data [1] to demonstrate and verify the performance of our UCB-FDF policy as well as the impacts of delayed feedback on real-world scenarios. We also verify our theoretical analysis through various product categories and demonstrate the efficacy of our proposed UCB-FDF MAB policy.

The rest of the paper is organized as follows. In Section 2, we review the literature to put our work in comparative perspectives. In Section 3, we formulate our new MAB model that captures the three common phenomena. In Section 4, we present our UCB-FDF policy and analyze its performance.

---

[1] https://nijianmo.github.io/amazon/

Then, we present our experiment settings and results in Section 5. Due to space limitations, the proofs and part of experiemntal results are relegated to the appendix.

## 2 RELATED WORK

In this section, we provide a quick overview on three lines of research related to our work: i) bandits with delayed feedback, ii) bandits with random preferences, and iii) incentivized bandits.

**1) Bandits with Delayed Feedback:** Motivated by practical issues in the clinical trials, Eick (1988) was the first to introduce a two-armed bandit model with delayed responses, where the patients' survival time reports after the treatment are delayed. Recently, Joulani et al. (2013) provided a systematic study and showed that for delay $\tau$ with a finite expectation, the worst case regret scales with $O(\sqrt{KT \log T} + K\mathbb{E}[\tau])$, where $K$ is the number of arms. Meanwhile, Vernade et al. (2017) showed that stochastic MAB problems with delayed feedback have a regret lower bound $O(K \log T)$. However, this work assumed that the distribution of the random delay is arm-independent. In contrast, Joulani et al. (2013) considered arm-dependent delay distributions that have an upper bound of the maximum random delay. More recently, Manegueu et al. (2020) considered arm-dependent and heavy-tailed delay distributions, where only an upper bound on the tail of the delay distribution is needed, without requiring the expectation to be finite. Also, Lancewicki et al. (2021) studied the case where the delay distribution is reward-dependent, which implies that the random delay in each round may also depend on the reward received on the same round. However, most of these works on delayed bandits are based on the standard stochastic MAB framework. In contrast, we consider delayed feedback in incentivized bandit learning with self-reinforcing user preferences, which is a more appropriate model for real-world recommender systems than the standard stochastic MAB.

**2) Bandits with Random Preferences:** The impacts of random user preferences in e-commerce platforms have received increasing interest in several different areas in learning and economics. Existing works in (Agrawal et al., 2017; 2019) formulated the user preference variation given different product bundles by the multi-nomial logit model on top of the bandit learning framework and proposed a Thompson Sampling approach that achieves a worst-case regret bound of $O(\sqrt{NT} \log TK)$, where $N$ is the size of recommended arm bundle. With a different focus on preference modeling, Barabási & Albert (1999); Chakrabarti et al. (2006); Ratkiewicz et al. (2010) investigated the network evolution with "preferential attachment" that formulates the social behavior known as self-reinforcing preferences, among which the works in (Shah et al., 2018; Zhou et al., 2021) are the closet to our work. To our knowledge, Shah et al. (2018) was the first to consider self-reinforcing user preferences in bandit learning problems. Later, Zhou et al. (2021) incorporated self-reinforcing user preferences into the incentivized bandit learning framework. The key difference between these two works is that, in the model in (Shah et al., 2018), only one arm is revealed to users in each round, while in the model in (Zhou et al., 2021), all arms are revealed to users and users' arm selections are influenced by incentives. However, both of these works fall short in modeling online recommender systems in practice as they assume that an arm-sampling feedback is observable in the same timeslot when an arm is pulled. However, for most e-commerce recommender systems in practice, user feedbacks are often not immediately observable. As a result, the decision on which arm to pull next has to be made without some of the feedbacks from arm-pulling actions in the past.

**3) Incentivized Bandits:** To our knowledge, Frazier et al. (2014) was among the first to adopt incentive schemes into a Bayesian MAB setting. In their model, the agent seeks to maximize time-discounted total reward by incentivizing arm selections. Later, Mansour et al. (2015) studied the non-discounted reward setting. For the non-Bayesian setting, Wang & Huang (2018); Zhou et al. (2021) proposed policies that maximize the total non-discounted reward with bounded incentive costs. Bandits with budget (Guha & Munagala, 2007; Goel et al., 2009) also share some similarities with our work, where the agent takes actions under resource constraints that are either fixed or with a given growth rate bound. However, none of the aforementioned works considered the impacts of delayed feedback on the regret and incentive costs performances. Note that, due to the loss of information caused by delayed feedback, larger variances in the mean-reward estimations of the arms are inevitable. This implies that, in order to achieve a more accurate arm quality estimation under delayed feedbacks, a higher incentive cost is necessary.

## 3 SYSTEM MODEL

The system has a set of $K \geq 2$ arms denoted by $\mathcal{A} = \{1, \ldots, K\}$, and each arm $a$ follows a Bernoulli reward distribution $P_a$ with an unknown mean $\mu_a > 0$. The bandit time horizon has $T$ rounds. In each time step $t = 1, 2, \ldots, T$, a user arrives and chooses an arm $I_t$ to pull. Then, the user will receive a random reward feedback $X_t \sim P_{I_t}$. Both the arm selection $I_t$ and the feedback $X_t$ are observable to the agent. We use $T_a(t) \triangleq \sum_{i=1}^{t} 1_{\{I_i = a\}}$ to denote the number of times that arm $a$ is pulled up to time step $t$. We let $T_a(0) = 0, \forall a \in \mathcal{A}$. We assume that there is a unique best arm $a^* \in \mathcal{A}$ in the sense that $a^* = \arg\max_{a \in \mathcal{A}} \mu_a$ and let $\mu^* = \mu_{a^*}$. Also, we define $\Delta_a \triangleq \mu^* - \mu_a$ as the gap between the mean of the optimal arm and the mean of arm $a$.

### 3.1 DELAYED FEEDBACK MODELING

In this paper, we consider delayed feedback, i.e., when an arm $I_t$ is pulled at time step $t$, the corresponding Bernoulli reward $X_t$ is observed after a delay period $\tau_{I_t,t}$, i.e., the feedback $X_t$ is observed at time step $t + \tau_{I_t,t}$. Without loss of generality, we model the random delay time as a random variable $\tau_{a,t} \sim \mathcal{T}_a$, where the delay distribution $\mathcal{T}_a$ of arm $a$ is unknown to the agent.

We consider two settings of delayed feedback. We first consider i.i.d. delays $\{\tau_t\}_{t \leq T}$ across time and arms, i.e., the delay distributions are identical for all arms. Thus, we omit the arm index in the notations of delay feedback in this setting. Next, we generalize the delay modeling by allowing arm-dependent delay distributions, where the delay distributions are allowed to differ across arms. In both settings, we do not make further assumptions on the delay distributions, except that we only require a finite delay expectation. Note that we allow the support of the delays to be unbounded, i.e., an infinite delay time is possible in both settings. This models the practical scenarios in online recommender systems that some user feedbacks (e.g., ratings and reviews) may never be received.

Under delayed feedbacks, we denote the total number of missing feedbacks from arm $a$ up to a time step $t$ as $D_a(t) \triangleq \sum_{s=1}^{t} 1_{\{s + \tau_{a,s} > t\}}$. We let $D_a^*(t) = \max_{1 \leq s \leq t} D_a(s), \forall a \in \mathcal{A}$ as the maximum total number of delayed feedback for arm $a$ up to time $t$. Note that $D_a^*(t) = 0, \forall a \in \mathcal{A}$ corresponds to the non-delayed setting. In this case, $T_a(t)$ denotes the total number of pulling times of arm $a$ up to time $t$. At each time step $t$, the agent observes a set of time-stamped feedback denoted by $S_t \subset \mathbb{N} \times \{0, 1\}$. In the set $S_t$, each element is a pair of time index and a Bernoulli reward value, and the time index is the time step when the corresponding reward is observable. Note that in this model, by observing the set $S_t$, the agent is aware of the information of both the time step when the feedback is received, and the arm that generated the feedback. We denote the total reward generated by arm $a$ up to time $t$ as $S_a(t) \triangleq \sum_{s=1}^{t} X_s \cdot 1_{\{I_s = a, s + \tau_{a,s} \leq t\}}$, and let $S_a(0) = 0, \forall a \in \mathcal{A}$.

### 3.2 USER PREFERENCES AND INCENTIVE IMPACT MODELING

In this paper, we assume that the arrival at time $t$ has a non-zero probability $p_a(t) \in (0, 1)$ to pull each arm $a \in A$. We note that $p_a(t)$ can also be thought of as the user's preference rate of arm $a$, and $\sum_{a \in \mathcal{A}} p_a(t) = 1, \forall t \leq T$. We adopt the widely accepted multinomial logit model in the economics literature(Bawa & Shoemaker, 1987) to model arm $a$'s preference rate at time step $t$ as follows:

$$p_a(t) = \frac{f\big(S_a(t-1) + \theta_a\big)}{\sum_{i \in A} f\big(S_i(t-1) + \theta_i\big)}, \tag{1}$$

where $f(\cdot) : \mathbb{R} \to (0, +\infty)$ is a feedback function that is increasing, and $\theta_a > 0$ denotes a fixed initial preference bias of arm $a$. We note that the preference rate modeling in (Zhou et al., 2021) is also based on the multinomial logit model, which appears to be in the same form as in (1). However, the key difference between our preference model in (1) and that in (Zhou et al., 2021) is that the accumulative award information $S_i(t-1)$ in (1) accounts for reward information that can only be observed up to time $t$. In other words, $S_i(t-1)$ in (1) is affected by feedback delays. In fact, the preference model in (Zhou et al., 2021) can be viewed as a special case of our model with zero delay.

Since the arriving users select arms based on preferences, while the agent aims to maximize the total reward in the long run, there exists a general difference between users' arm preferences and agent's intended arm selection. To induce users to pull arms following the agent's goal, the agent needs to intervene users' arm pulling by offering incentives on its desired arm, so as to increase the user'

preference of pulling the arm. That is, the agent incentivizes arm $I_t'$ at time step $t$ so that $p_{I_t'}(t)$ increases accordingly. Note that when $p_{I_t'}(t)$ increases, the preference rates on the other arms will decrease since $\sum_{a \in \mathcal{A}} p_a(t) = 1, t \leq T$. In this paper, we adopt the "coupon effect" model, which is widely used in the economics and marketing literature (Bawa & Shoemaker, 1987). Specifically, we consider a fixed incentive $b$ in each time step and denote the time-dependent incentive impact as $g(b, t)$. Then, the posterior preference rates of the arms with incentive $b$ are updated as follows:

$$\hat{p}_i(t) = \begin{cases} \dfrac{p_i(t) + g(b, t)}{1 + g(b, t)}, & i = a, \\ \dfrac{p_i(t)}{1 + g(b, t)}, & i \neq a. \end{cases} \tag{2}$$

We remark that the definition of the posterior preference update in (2) also follows from the multi-nomial logit model, which is widely used to model user preferences and their variations in bandit field (Chen & Wang, 2017; Avadhanula, 2019; Dong et al., 2020; Zhou et al., 2021). Based on the defined posterior preference, as incentive impact $g(b, t)$ increases to infinity (either the incentive value $b$ increases to infinity or the users are more sensitive to incentives as time goes by), the user preference will be induced to pulling the agent's desired arm $a$ with probability one. For further detailed interpretations of the incentive impact function $g(b, t)$, we refer readers to the literature (e.g., (Zhou et al., 2021)). Note also that, due to the random user behaviors, it is possible that $I_t' \neq I_t$, i.e., the arm that the agent incentivizes is not the one that a user pulls eventually. We define the accumulative incentive up to time step $t$ as $B_t \triangleq \sum_{s=1}^{t} b_t$, where $b_t \in \{0, b\}, \forall t \leq T$, denotes the agent's binary decision whether to offer incentive $b$ at time step $t$.

## 3.3 Regret Modeling

As in most bandit learning problems, the goal of the agent is to maximize the total expected reward $\mathbb{E}\left[\sum_{a \in \mathcal{A}} S_a(T)\right]$ in the long run. Toward this end, we need the notion of the oracle incentivized policy, where in hindsight, the agent is aware of the optimal arm $a^*$ and can always offer an infinite amount of payments to users with feedback being observable immediately, so that the posterior preference rate of arm $a^*$ is always infinitely close to one. As a result, the expected accumulative reward generated under the oracle policy up to time $T$ is $\mathbb{E}[S_{a^*}(T)] = \mu^* \cdot T$. However, since the optimal arm $a^*$ is unknown to the agent, the goal of the agent is to maximize the total expected reward $\mathbb{E}[\Gamma_T]$ in the long run by designing an incentivized policy with low accumulative incentive in the presence of self-reinforcing preferences and feedback delay. Similar to conventional MAB, we measure the performance gap between our accumulative reward against that of the oracle policy, which is denoted by regret $R_T$. The expected (pseudo) regret is defined as follows:

$$\mathbb{E}[R_T] = \mu^* \cdot T - \mathbb{E}\left[\sum_{a \in \mathcal{A}} S_a(T)\right].$$

In this paper, our goal is to minimize $\mathbb{E}[R_T]$ with low expected accumulative payment $\mathbb{E}[B_T]$, i.e., sub-linear growth rate regarding time horizon $T$. It is clear that any policy with bounded payment cannot outperform the oracle policy. Thus any expected regret defined by comparing with bounded-payment policy is upper bounded by our regret.

## 4 Bandit Policy Design and Performance Analysis

In this section, we first present the general version of the UCB-FDF policy that works with any delay distributions, where we upper bound the delay impact on the regret and incentive costs. Based on this general result, we then study the regret and incentive costs performance of UCB-FDF under the assumptions of 1) i.i.d. feedback delay across arms/times and 2) arm-dependent delay distributions. In both cases, we denote the total number of missing feedbacks over all arms by $D(t) \triangleq \sum_{a \in \mathcal{A}} D_a(t)$, and denote the maximum number of missing feedbacks during the first $t$ time steps by $D^*(t) \triangleq \max_{1 \leq s \leq t} D(s)$. For arm $a$ at time step $t$, we denote the number of its pulling times whose feedback is observed by $T_a'(t) = T_a(t) - D_a(t)$, and denote the maximum mean gap by $\Delta^* = \max_{a \in \mathcal{A}} \Delta_a$. At time step $t$, we denote the sample mean estimation (due to delayed feedbacks) of arm $a$ by $\hat{\mu}_a(t) = S_a(t)/T_a'(t)$. Our UCB-FDF policy is illustrated in Algorithm 1.

---

**Algorithm 1** The UCB-Filtering-with-Delayed-Feedback Policy (UCB-FDF).

---

**Require:** Time horizon $T$ and incentive payment $b$, the confidence interval of arm $a$ at time step $t$ defined as $c_a(t) = \sqrt{\ln T / (2T'_a(t))}$.

1: **Initialization:** Incentivize pulling the arms satisfying $T'_a(t) = 0$ with incentive payment $b$ until $\min_{a \in \mathcal{A}} T'_a(t) \geq 1$. Let set $\mathcal{U} = \mathcal{A}$. Mark current time as $t_0$.

2: **Exploration Phase:** While $|\mathcal{U}| > 1$, remove all the arms from set $\mathcal{U}$ satisfying $\hat{\mu}_a(t) + c_a(t) \leq \max_{i \neq a, i \in \mathcal{U}} (\hat{\mu}_i(t) - c_i(t))$ if there is any, then incentivize pulling arm $a \in \arg\min_{i \in \mathcal{U}} T'_a(t)$ with payment $b$. If $|\mathcal{U}| = 1$, let arm $\hat{a}^* = \{a : a \in \mathcal{U}\}$ and mark current time as $t_1$.

3: **Exploitation Phase:** Incentivize pulling arm $\hat{a}^*$ with payment $b$ until it dominates: $S_{\hat{a}^*}(t) \geq \sum_{a \neq \hat{a}^*} (S_a(t) + D_a(t))$. Mark current time as $t_2$.

4: **Self-Sustaining Phase:** Users pull arms based on their own preferences until time $T$.

---

UCB-FDF policy contains three phases: an incentivized exploration phase, an incentivized exploitation phase, and a self-sustaining phase. UCB-FDF policy tackles feedback delays in the following two key aspects: (i) correcting the sample mean estimate of arms by only considering the number of pulling times that have observed feedback, (ii) setting the length of the exploitation phase in such a way that the outstanding rewards do not harm the emergence of "dominance" (i.e., one arm receiving at least half of the rewards) of the sampled optimal arm. Subsequently, these two aspects also influence the regret and incentive. In order to have enough arm exploration with an unbiased sample mean estimate, the loss of counted number of pulling times necessitates a carefully designed exploration phase that incentivizes the pulling of the least informed arm $a \in \arg\min_{i \in \mathcal{U}} T'_a(t)$ under delayed feedbacks. Similarly, the delay-based dominance threshold (i.e., $S_{\hat{a}^*}(t) \geq \sum_{a \neq \hat{a}^*} (S_a(t) + D_a(t))$ in Step 3 of Algorithm 1) guarantees the dominance of sampled optimal arm, while also accounts for a longer exploitation phase to mitigate the delayed feedback effect. We now analyze the upper bounds of the pseudo regret and expected incentive of the UCB-FDF policy.

**Lemma 1.** (UCB-Filtering-with-Delayed-Feedback) *Given a fixed time horizon $T$, if $g(b,t) > 1$, and $f(x) = \Theta(x^\alpha)$ with $\alpha > 1$[2], then the pseudo regret of Algorithm 1 $\mathbb{E}[R_T]$ is upper bounded by*

$$\mathbb{E}[R_T] \leq \sum_{a \neq a^*} \frac{8\Delta_a(g(b,1) - 1) + 8\Delta^*}{(g(b,1) - 1)\Delta_a^2} \ln T + \frac{g(b,1)\Delta^*(\mathbb{E}[D^*(T)] + 4K)}{g(b,1) - 1},$$

*with the expected payment $\mathbb{E}[B_T]$ upper bounded by*

$$\mathbb{E}[B_T] \leq b \cdot \frac{2g(b,1) + 1}{g(b,1) - 1} \left[ \frac{8\ln T}{\Delta_{min}^2} + \sum_{a \neq a^*} \frac{8\ln T}{\Delta_a^2} + \mathbb{E}[D^*(T)] + 4K \right].$$

**Remark 1.** The UCB-FDF policy achieves a sub-linear total incentive cost by leveraging the property of self-reinforcing preference. We can show that as long as the self-reinforcing preference function $f(x)$ satisfies the condition $f(x) = \Theta(x^\alpha)$ with $\alpha > 1$, then "monopoly" happens with probability one (i.e., the scenario where only one arm has positive probability to be pulled, thus this particular arm is the only preferred arm). A natural incentivizing policy is to incentivize sampled optimal arm until it achieves monopoly. However, the key challenge here is that the onset of monopoly could take infinite time steps, which implies linear total incentive. Moreover, self-reinforcing property is not merely disrupting the system from converging to the optimal arm. The key idea in our UCB-FDF policy design is that under the condition of the self-reinforcing preference function $f(\cdot)$, after one arm establishes its dominance (i.e., the arm $a$ generates at least half of the current total reward), it will have exponentially increasing probability to beat other arms and achieve monopoly. More importantly, we can show that the onset of arm dominance takes sub-linear times, thus allowing us to achieve sub-linear total incentive costs.

**Remark 2.** The feedback delay affects the observation of arm dominance, since the missing reward information from suboptimal arms, if not compensated carefully, can potentially destroy the dominance status of the optimal arm. Thus, to guarantee dominance of the optimal arm, a longer exploitation phase is necessary, and thus a large total incentive is required.

---

[2]The notation $\Theta()$ in this paper is defined as that, if $f(x) = \Theta(g(x))$, then there exist $x_0$ and two constants $C_1, C_2 > 0$, such that $C_1 g(x) \leq f(x) \leq C_2 g(x)$ for all $x \geq x_0$.

The existence of delays in our MAB model introduces an additive term $\Theta(\mathbb{E}[D^*(T)])$ in both regret and incentive costs, which is dependent on the maximum accumulated delayed feedback up to time horizon $T$. Based on Lemma 1, in what follows, we will analyze the upper bounds of the expected maximum accumulated delayed feedback under different assumptions on delay distributions.

## 4.1 ARM-INDEPENDENT DELAY WITH A FINITE EXPECTATION

We now analyze the delay impact under our first assumption. In the arm-independent case, we consider an i.i.d. sequence $\{\tau_t\}_t$ of random delay regarding time step $t \leq T$. We do not make any assumption on the shape of the delay distribution, except that we only assume a finite expectation $\mathbb{E}[\tau_1]$. Thus, an infinite random delay is possible under this assumption, implying some feedbacks may never be observed by the agent. Our results show that under this assumption, we can still achieve similar orders of the regret and incentive costs growth rates, since the key fact is that we can upper bound the expected number of such unexpectedly large random delays for every time step $t$.

Existing works (e.g., (Joulani et al., 2013)) provided a systematic study on the delay effect on the partial monitoring problem with side information, including the stochastic problems. Although these works only considered the classic stochastic MAB, they share some similarities with our work in that their analysis of delay effects also leveraged the maximum number of missing feedbacks during the first $t$ time steps $D^*(t)$. However, since our UCB-FDF policy has a different structure compared to these works on delayed stochastic MAB, their delay analysis is not applicable to our policy. Next, we restate a result in (Joulani et al., 2013), which will be useful in our analysis.

**Lemma 2** (Lemma 2 in Joulani et al. (2013)). *Assume $\{\tau_1, \ldots, \tau_t\}$ is a sequence of i.i.d. random variables with finite expected value, and let $B(t, s) = s + 2\log t + \sqrt{4s \log t}$. Then, it holds that*

$$\mathbb{E}[D^*(t)] \leq B(t, \mathbb{E}[\tau_1]) + 1.$$

**Theorem 3.** (Arm-Independent Delay) *Under i.i.d. delays with a finite expectation and the conditions of Lemma 1, the pseudo regret of Algorithm 1 $\mathbb{E}[R_T]$ is upper bounded by*

$$\left[\frac{2g(b,1)\Delta^*}{g(b,1)-1} + \sum_{a \neq a^*} \frac{8\Delta_a\big(g(b,1)-1\big) + 8\Delta^*}{\big(g(b,1)-1\big)\Delta_a^2}\right] \ln T + \frac{g(b,1)\Delta^*\big(\sqrt{4\mathbb{E}[\tau_1]\ln T} + \mathbb{E}[\tau_1] + 4K + 1\big)}{g(b,1)-1},$$

*with the expected payment $\mathbb{E}[B_T]$ upper bounded by*

$$b \cdot \frac{2g(b,1)+1}{g(b,1)-1}\left[\left(2 + \frac{8}{\Delta_{min}^2} + \sum_{a \neq a^*} \frac{8}{\Delta_a^2}\right)\ln T + \sqrt{4\mathbb{E}[\tau_1]\ln T} + \mathbb{E}[\tau_1] + 4K + 1\right].$$

We note that the gap summation of arms $\sum_{a \neq a^*} \Delta_a$ plays an important role in both regret and total incentive. As arm gaps getting smaller, it is more difficult to distinguish the optimal arm from others. Thus, a longer exploration phase is required to conduct enough sampling, which implies a larger regret and a larger total incentive costs. On the other hand, the feedback delay causes additive terms in both regret and incentive costs in terms of the expected delay $\mathbb{E}[\tau_1]$, and the delay impact can be upper bounded as long as the expected delay $\mathbb{E}[\tau_1]$ is no larger than time horizon $T$.

## 4.2 ARM-DEPENDENT DELAY WITH FINITE EXPECTATIONS

Now, we further relax the assumption on the delay to allow *arm-dependent* delays. In this case, the delay has two key impacts on the system: (i) for each arm, there is a different real-time information loss when estimating the sample mean, (ii) for the whole arm set, different scales of delay cause an *uneven* arm estimation, which results in a larger risk of the elimination of the optimal arm in the UCB-based exploration step. We formally state our arm-dependent delay assumption as follows:

**Assumption 1.** *The delays of arm $a \in \mathcal{A}$ form an independent delay sequence $\{\tau_{a,t}\}$, where each element is a random variable satisfying $\tau_{a,t} \sim \mathcal{T}_a$, with a finite expectation $\mathbb{E}[\tau_{a,1}] < +\infty$, $\forall a \in \mathcal{A}$.*

Under Assumption 1, we show a more general result on the upper bound of $\mathbb{E}[D^*(t)]$ as follows:

**Lemma 4.** *Under Assumption 1, given a finite number of arms $K > 0$, it holds that*

$$\mathbb{E}[D^*(t)] \leq \sum_{a \in A} 2\mathbb{E}[\tau_{a,1}] + 3K \log \frac{t}{K}.$$

The result in Lemma 4 implies larger upper bounds of the regret and incentive, due to the existence of the pre-log factor $K$. This is a consequence of the situation where, as we consider arm-dependent delay distributions, the worst case could be evenly distributed expected delays $\mathbb{E}[\tau_{a,1}]$ of arm $a$ with respect to time horizon $T$. Formally, we state the upper bounds of regret and incentive as follows:

**Theorem 5.** (Arm-Dependent Delay) *Under Assumption 1 and the conditions of Lemma 1, the pseudo regret of Algorithm 1* $\mathbb{E}[R_T]$ *is upper bounded by*

$$\sum_{a \neq a^*} \frac{8\Delta_a\big(g(b,1)-1\big) + 8\Delta^*}{\big(g(b,1)-1\big)\Delta_a^2} \ln T + \frac{g(b,1)\Delta^*\big(3K \ln \frac{T}{K} + \sum_{a \in \mathcal{A}} 2\mathbb{E}[\tau_{a,1}] + 4K\big)}{g(b,1)-1},$$

*with the expected payment* $\mathbb{E}[B_T]$ *upper bounded by*

$$b \cdot \frac{2g(b,1)+1}{g(b,1)-1}\bigg[\bigg(\frac{8}{\Delta_{min}^2} + \sum_{a \neq a^*} \frac{8}{\Delta_a^2}\bigg) \ln T + 3K \ln \frac{T}{K} + \sum_{a \in \mathcal{A}} \mathbb{E}[\tau_{a,1}] + 4K\bigg].$$

Similar to the results under the i.i.d. delay assumption, we can still upper bound the regret and incentive by an logarithmic growth rate $O(\log T)$ under arm-dependent delay. This implies that even under the weak delay assumption where only finite expectation is needed, UCB-FDF can estimate arms without too much bias, and finally achieve logarithmic regret with logarithmic incentive costs.

## 5 EXPERIMENTAL RESULTS

In this section, we first introduce our experiment setting and the dataset, then illustrate our experimental results. Due to the space limit, the full experimental results are provided in Appendix **??**.

### 5.1 EXPERIMENTAL SETUP

**1) System Parameters:** We conduct experiments under two different delay settings. The system parameters are set as follows: a three-armed model with *Arm1* being the optimal arm, and the initial preference bias $\theta = [1, 5, 5]$, i.e., the optimal arm has the least initial bias. We choose a three-armed model since large arm set requires a proportional large time horizon to distinguish optimal arm, while in the public Amazon Review Data, the amount of reviews for most products is limited (no more than 3,000 for each product). The self-reinforcing preference function is chosen as $f(x) = x^\alpha$ with $\alpha = 2$. The constant incentive for each time step is set as $b = 1.5$ with an incentive impact function $g(b, t) = b$. For the delay distribution, we use normal distributions in both assumption setting, as normal distributions have an infinite support $x \in \mathbb{R}$. Under the arm-independent delay setting, we choose the delay distribution as $\tau_t \sim N(10, 2)$. Under the arm-dependent delay setting, we choose the delay distributions as $\tau_{1,t} \sim N(80, 2)$, $\tau_{2,t} \sim N(10, 2)$, and $\tau_{3,t} \sim N(10, 2)$ for Arms 1, 2, and 3, respectively. We only generate non-negative samples of delay under both assumptions.

| Product Category | Arm1(optimal) | Arm2 | Arm3 |
|---|---|---|---|
| Pet Supplies | 0.773 | 0.656 | 0.626 |
| Electronics | 0.757 | 0.605 | 0.617 |
| Home and Kitchen | 0.875 | 0.588 | 0.673 |
| Books | 0.915 | 0.551 | 0.706 |

Table 1: Means of products (arms) in different categories.

**2) Dataset:** We use **Amazon Review Data** (Ni et al., 2019) to provide a practical learning environment. The Amazon Review Data includes 233 million customer reviews (ratings, posting times) for 29 product categories. In the experiment, we select three products to serve as the arms that have the largest number of reviews in category Pet Supplies, Electronics, Home and Kitchen, and Books, respectively. For each product (arm), we leverage the `rating` and `unixReviewTime` information in each review, and the total number of reviews is 3,000 for each product. The range of ratings in Amazon Review Data is the discrete set $\{1, 2, 3, 4, 5\}$. We convert the rating values to binary by setting the rating values 1 and 2 as 0 and the rating value 4 and 5 as 1, and the reviews with rating

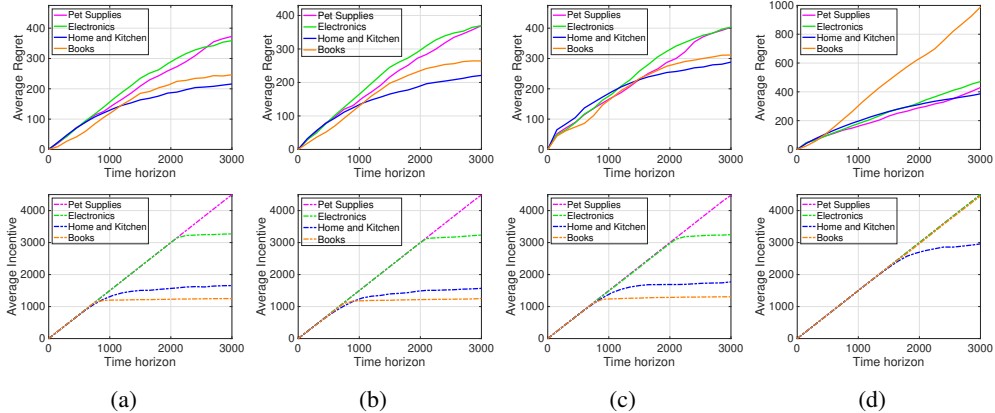

Figure 1: The performance of policy UCB-FDF in the face of no delay in (a), the performance of policy UCB-FDF in the face of arm-independent delay in (b), the performance of policy UCB-FDF in the face of arm-dependent delay in (c), and the performance of policy UCB-List in the face of arm-dependent delay in (d).

value 3 are removed. For each product, the binary review ratings are sorted by `unixReviewTime`, so the ratings come in real-world order in the experiment. We summarize the mean values of the products by their Bernoulli ratings in the four selected categories, as shown in Table 1.

## 5.2 RESULTS AND DISCUSSIONS

**1) Results:** The experiment results are illustrated in Figure 1. (a) shows the average regret and incentive trends with policy UCB-FDF under setting with no delay. (b) and (c) show the average regret and incentive trends with policy UCB-FDF under settings with arm-independent delays and arm-dependent delays, respectively. In Figures (c) and (d), we compare the performances with policy UCB-FDF and baseline policy UCB-List (Zhou et al., 2021). Specifically, Figure (d) shows the performance under policy UCB-List in the face of arm-dependent delays that is the same as that in (c). Each curve is constructed by regret or incentive values with different time horizons from $T = 150$ to $T = 3000$, incremented by $150$. Each node value in curves are averaged by 100 trials.

**2) Discussion:** Comparing (a) with (b) and (c) in Figure 1, we can observe the delay impact on regret and total incentive, that both the regret and total incentive are increased due to the delayed feedback. Comparing (c) and (d) in Figure 1, we observe that under the bandit instances in the face of same delayed feedback, our policy UCB-FDF reaches sub-linear growth rate in both regret and total incentive, except the total incentive in category Pet Supplies, since it may require more time steps to converge while our data is limited, while the policy UCB-List cannot guarantee sub-linear growth rate for both regret and total incentive.

## 6 CONCLUSION

In this work, we proposed a practical bandit model that considers the joint effect of the incentive impact, delayed feedback and self-reinforcing user preferences in real-world recommender systems. We proposed a UCB-FDF policy that achieves logarithmic growth rates of pseudo regret and total incentive costs for a fixed time horizon $T$. We also analyzed how different delay assumptions influence the regret and incentive costs. Specifically, we considered arm-independent delays and arm-dependent delays with weak assumption that only requires a finite expectation. The evaluations with real-world customer review data showed the effectiveness of our UCB-FDF policy in achieving sub-linear regret while spending only sub-linear total incentive costs under delayed feedback.

### ACKNOWLEDGMENTS

This work has been supported in part by NSF grants CAREER CNS-2110259, CNS-2112471, CNS-2102233, CCF-2110252, and a Google Faculty Research Award.

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
