# OpenReview forum: "Bandit Learning with Joint Effect of Incentivized Sampling, Delayed Sampling Feedback, and Self-Reinforcing User Preferences"
_ICLR.cc/2022/Conference — ICLR 2022 Poster_

### Official Review · Reviewer_8BXC · 2021-11-01

**Correctness:** 4
**Technical Novelty And Significance:** 3
**Empirical Novelty And Significance:** 2
**Recommendation:** 6
**Confidence:** 3

**Main Review:**

Pros:
1.	The motivation for introducing the three factors is explained very clearly. The description of Algorithm 1 and the intuition behind each step in the policy are also well-written.
2.	The comparisons with related works given in Section 2 are helpful.
3.	The theoretical results, to my understanding, are derived under fairly general assumptions. The distinctions between arm-independent versus arm-dependent delays are useful to see.

Cons:
1.	The experiment evaluation is insufficient in my opinion. In terms of verifying the provided regret bounds, I find the presented experimental results somewhat narrow. Given that the derived bounds contain many problem-specific parameters, such as arm number $K$, delay expectation $E[D*(T)]$, I think additional experiments with different setups would strengthen the empirical results. For example, would it be possible to replace the normal delay distributions with other distributions? What the results would look like at different self-reinforcing preference function and incentive impact function?
Additionally, I would expect the experiments to provide some empirical evidence for the benefits of considering these additional factors. For example, have the authors considered experiment comparisons between UCB-FDF on the new MAB framework versus a policy on the standard stochastic MAB framework?
2.	I briefly checked Zhou et al. 2021, which already considered two out of the three new factors. To my understanding, the main contribution from this paper is the additional factor of delayed feedback in the MAB framework. While I agree with the argued usefulness of allowing feedback delays, I am not sure how much novelty, or what are the new ideas needed, in the design of the UCB-FDF policy and the theoretical analysis. Please correct me if I were wrong: it appears that Algorithm 1 differs with Policy 2 in Zhou et al. 2021 only in the ‘Exploitation phase’ to include delay $D_a(t)$ into the dominance criterion. Is it the obvious choice, or is there more sophisticated argument underneath?

Some minor suggestions:
1.	The mathematical formalization for self-reinforcing preference is a little difficult to find in the paper.
2.	I believe $\Theta()$ is never formally defined in the paper. This may be a standard expression, but I think giving its definition would still be helpful.
3.	Would it be possible to mark $t_1, t_2$ to indicate the lengths of each phase when plotting the experiment results over the selected time horizon $T$?

Questions during rebuttal period:
Please address and clarify the cons above. In particular, could the authors highlight what are the theoretical novelty of the proposed framework, policy and theoretical bound derivations, in comparison to the cited works?


**Summary Of The Paper:**

This paper proposes a multi-armed bandit (MAB) framework with three realistic considerations: incentivized sampling, delayed feedback and self-reinforcing preferences. The paper proposes a ‘UCB-Filtering-with-Delayed-Feedback’ (UCB-FDF) policy for the new MAB framework. For general feedback delays with bounded expectations, the authors showed that the delayed sampling feedback has additive penalty on regret and incentive costs, then utilized this key fact to derive that the UCB-FDF policy achieves logarithmic regret and incentive cost in the new MAB framework. The theoretical bounds are verified by experiments on instances with 3 arms using Amazon Review Data.

**Summary Of The Review:**

I vote for 6: marginally above the acceptance threshold. The paper undertakes the ambitious goal of incorporating the joint effects of three new factors into a more realistic MAB framework. The proposed UCB-FDF policy is shown to achieve desirable logarithmic regrets without excessive incentive spending, and the theoretical results can apply to quite general delay distributions. Although I think the paper is well-written and the results are useful, I find the current version lacking in two main aspects. First, the experiments are limited: in addition to verifying the theoretical bounds, the experiments could also provide more insights about the UCB-FDF policy and/or compare with alternative MAB frameworks to demonstrate the advantages of incorporating the new factors. Second, the technical novelty and significance should be highlighted in a more clear way.

---

> ### Author Response · Authors · 2021-11-19
> **Author Response**
>
> We sincerely thank the reviewer's constructive comments and valuable insights, which help improve the quality of our work significantly. We have carefully revised our paper according to your comments and suggestions. Please see our revised submission, where we have highlighted all major changes in **Blue color**. Please also see our point-to-point responses as follows:
>
> > **Your Comment:** 1. The experiment evaluation is insufficient in my opinion. In terms of verifying the provided regret bounds, I find the presented experimental results somewhat narrow. Given that the derived bounds contain many problem-specific parameters, such as arm number $K$, delay expectation $E[D^*(T)]$, I think additional experiments with different setups would strengthen the empirical results. For example, would it be possible to replace the normal delay distributions with other distributions? What the results would look like at different self-reinforcing preference function and incentive impact function? Additionally, I would expect the experiments to provide some empirical evidence for the benefits of considering these additional factors. For example, have the authors considered experiment comparisons between UCB-FDF on the new MAB framework versus a policy on the standard stochastic MAB framework?
>
> **Our Response:** Thanks for the suggestions! In the paper revision, we added three groups of experiments: i) baseline comparison with existing policy UCB-List in Zhou et al. 2021, where similar bandit setting is considered except delay impact; ii) delay distribution comparison under different delay distributions and distribution parameters; iii) comparison with different system parameters, in particular, we compare performance under different feedback functions $f(\cdot)$ and incentive impact functions $g(b,t)$. For the new experiments, please refer to Appendix II in our revised paper.
>
> > **Your Comment:** 2. I briefly checked Zhou et al. 2021, which already considered two out of the three new factors. To my understanding, the main contribution from this paper is the additional factor of delayed feedback in the MAB framework. While I agree with the argued usefulness of allowing feedback delays, I am not sure how much novelty, or what are the new ideas needed, in the design of the UCB-FDF policy and the theoretical analysis. Please correct me if I were wrong: it appears that Algorithm 1 differs with Policy 2 in Zhou et al. 2021 only in the ‘Exploitation phase’ to include delay $D_a(t)$ into the dominance criterion. Is it the obvious choice, or is there more sophisticated argument underneath?
>
> **Our Response:** Thanks for your question. Yes, the key difference in this work compared with (Zhou et al. 2021) is the delayed feedback. We note that this difference is by no means trivial. Specifically, the delayed feedback effect on top of the bandit model (Zhou et al. 2021) induces the following new challenge: if a bandit policy does not handle the delayed feedback appropriately and makes decisions with outdated information, then sub-optimal arms will have the higher chance to collect more rewards, which would in turn lead to the higher user preferences on sub-optimal arms. This self-reinforcing preferences on sub-optimal arms will lead to further sub-sampling on the optimal arm, which forms a *vicious cycle* and could even result in a linear regret if a sub-optimal arm reaches *monopoly*. Thus, the impacts of delayed feedback render our bandit model far more complex than that in (Zhou et al. 2021). Unfortunately, the proposed policy in (Zhou et al.2021) assumed immediate feedback, thus cannot address this challenge.
>
> Specifically, for our bandit model that suffers delayed feedback, the policy in (Zhou et al. 2021) will have biased estimation on arms due to the outdated reward feedback information. This may render sub-optimal arms having the chance to collect more rewards and thus increases the regret. Also, the exploitation phase cannot guarantee the dominance of the empirical best arm due to delayed reward feedback, which could lead to a larger regret, or even linear regret when a sub-optimal arm reaches monopoly.
>
> To Handle the above challenges, our policy has two major improvements: (i) in the exploration phase, we adjust the empirical mean estimation $\hat\mu_a(t)$ for arms as well as the confidence interval $c_a(t)$ in the presence of delayed reward feedback, so that the arm estimation remained unbiased (see Step 2 in Algorithm 1), although with larger variances due to the feedback delay; (ii) we show that the delay impact at time $t$ can be gathered in terms of the total number of missing feedbacks at time $t$, and the summation of missing feedbacks from all the empirical sub-optimal arms is exactly the number of rounds that we need to prolong the exploitation phase so as to guarantee the dominance of the empirical best arm. All these results are new in the literature.
>
> Please continue to see our next response below.

---

> > ### Author Response · Authors · 2021-11-19
> > **Author Response (Continued)**
> >
> > > **Your Comment:** 3. The mathematical formalization for self-reinforcing preference is a little difficult to find in the paper.
> >
> > **Our Response:** Given that the mathematical formalization for self-reinforcing preferences have been discussed at-length in (Zhou et al. 2021), to avoid repetition and due to space limitation, we have omitted some of the modeling details of self-reinforcing user preferences and instead referred readers to (Zhou et al. 2021).
> >
> > > **Your Comment:** 4. I believe $\Theta()$ is never formally defined in the paper. This may be a standard expression, but I think giving its definition would still be helpful.
> >
> > **Our Response:** Thanks for the suggestion. Here, $\Theta()$ follows from the standard family of Bachmann–Landau asymptotic notations, meaning "both upper and lower bounded by". More precisely, $f(x) = \Theta(g(x))$ means that there exist $x_0$ and two constants $C_1,C_2>0$, such that $C_1 g(x) \leq f(x) \leq C_2 g(x)$ for all $x \geq x_0$. We have added this clarification in the revised version (see Page 6 in the footnote).
> >
> > > **Your Comment:** 4. Would it be possible to mark $t_1$, $t_2$ to indicate the lengths of each phase when plotting the experiment results over the selected time horizon $T$?
> >
> > **Our Response:** Thanks for the suggestion! Yes, we added a table that lists all the average $t_1$s and $t_2$s over different time horizons and different delay distributions (see Appendix II, Table 2). We also provide part of the table (in the case $T=3000$) below for a quick glance:
> >
> > |            | No delay | $N(40, 5)$ | $Exp(0.01)$ | Arm-dependent |
> > | ---------- | :------- | ---------- | ----------- | ------------- |
> > | $\bar t_1$ | 818      | 796        | 927         | 800           |
> > | $\bar t_2$ | 835      | 847        | 1051        | 854           |
> >
> > > **Your Comment:** 5. In particular, could the authors highlight what are the theoretical novelty of the proposed framework, policy and theoretical bound derivations, in comparison to the cited works?
> >
> > **Our Response:** Thanks for your suggestion. Here, we would like to further clarify our novelty in terms of proposed framework, policy design, and theoretical regret bound analysis compared to existing works:
> >
> > * *Proposed Framework*: The combination of delayed feedback with self-reinforcing user preferences induces a non-trivial challenge: if a bandit policy does not handle the delayed feedback appropriately and makes decisions with outdated information, then sub-optimal arms will have the higher chance to collect more rewards, which would in turn lead to the higher user preferences on sub-optimal arms. This self-reinforcing preferences on sub-optimal arms will lead to further sub-sampling on the optimal arm, which forms a *vicious cycle* and could even result in a linear regret if a sub-optimal arm reaches *monopoly*. This challenge has yet been considered in existing bandit literature.
> > * *Policy Design:* Please refer to Our Response to Your Comment 2.
> > * *Theoretical Regret Analysis:* To upper bound both regret and total incentive, our policy UCB-FDF induces the theoretical analysis to $D^*(t)$, i.e., the maximum accumulated delayed feedback up to time $t$. The upper bound of $D^*(t)$ under two different delay assumptions in the face of self-reinforcing user preferences is yet considered in the existing bandit literature.

---

> > > ### Comment · Reviewer_8BXC · 2021-11-23
> > > **Thank you for your response**
> > >
> > > I appreciate the authors' efforts in providing the detailed responses to my questions as well as adding the new experiment results. The experiment section is greatly strengthened with the new contents, which provide useful insights about how the delay distributions and incentive function affect the results. The responses also clarified the paper's contribution in comparison to literature. I would keep my score unmodified as I view the paper's contributions useful but somewhat narrow.

---

### Official Review · Reviewer_mJjk · 2021-11-03

**Correctness:** 3
**Technical Novelty And Significance:** 2
**Empirical Novelty And Significance:** 3
**Recommendation:** 5
**Confidence:** 4

**Main Review:**

Major comments:

1.	This paper is largely motivated by Zhou et al. 2021, which incorporated self-reinforcing user preferences into the incentivized bandit learning framework. The key difference is that this work considers the delay effect. That is, the accumulative award information accounts for reward information that can only be observed up to time t, while Zhou et al. 2021 can observe the feedback immediately. Could you elaborate more on the technical difficulty when you consider the delay effect, compared to the work Zhou et al. 2021?

2.	In bandits with delayed feedback [Pike-Burke et al. 2018], their Theorem 2 illustrates that the regret bound has an additional term log(1/\Delta_j) E[\tau], which does not depend on T. However, in your Theorem 3, it has an additional term \sqrt{4 E[\tau_1] ln T}, which has the dependence on T. Why is that? Can it be improved? Could you derive the lower bound and close the gap on the dependence of the delay period?

3.	In Assumption 1, it assumes that the delays of arm a follows an independent delay sequence {\tau_{a,t}}, where each element is a random variable satisfying \tau_{a,t}~T_a. Can the result be generalized to the setting where \tau_a,t follow different distributions when t varies?

4.	Why only the term g(b,1) shows up in the regret bound but not g(b,t) for t>1?

5.	Regarding the numerical experiments, how do you choose the self-reinforcing preference function f(x) and the incentive impact function g(b,t)? In addition, can you compare the regret between the setting with delay effect and that without the delay effect, and different delay distributions, through which to illustrate the influence of the delayed feedback?


**Summary Of The Paper:**

This paper considers a MAB framework with joint effect of incentivized sampling, delayed sampling feedback, and self-reinforcing user preferences. The framework considers delayed feedbacks to reflect a more practical setting where customer preferences among products are influenced and reinforced by historical feedbacks.

**Summary Of The Review:**

The problem this paper studied is defined in a clear way, but authors may need to emphasize the technical contribution that is built upon the existing work and demonstrate the tightness of the regret bound.

---

> ### Author Response · Authors · 2021-11-19
> **Author Response**
>
> We sincerely thank the reviewer's constructive comments and valuable insights, which help improve the quality of our work significantly. We have carefully revised our paper according to your comments and suggestions. Please see our revised submission, where we have highlighted all major changes in **Blue color**. Please also see our point-to-point responses as follows:
>
> > **Your Comment:** 1. This paper is largely motivated by Zhou et al. 2021, which incorporated self-reinforcing user preferences into the incentivized bandit learning framework. The key difference is that this work considers the delay effect. That is, the accumulative award information accounts for reward information that can only be observed up to time t, while Zhou et al. 2021 can observe the feedback immediately. Could you elaborate more on the technical difficulty when you consider the delay effect, compared to the work Zhou et al. 2021?
>
> **Our Response:** Thanks for your question. Yes, we agree that the key difference in this work compared with (Zhou et al. 2021) is the delayed feedback. We note that this difference is by no means trivial. Specifically, the delayed feedback effect on top of the bandit model (Zhou et al. 2021) induces the following new challenge: if a bandit policy does not handle the delayed feedback appropriately and makes decisions with outdated information, then sub-optimal arms will have the higher chance to collect more rewards, which would in turn lead to the higher user preferences on sub-optimal arms. This self-reinforcing preferences on sub-optimal arms will lead to further sub-sampling on the optimal arm, which forms a *vicious cycle* and could even result in a linear regret if a sub-optimal arm reaches *monopoly*. Thus, the impacts of delayed feedback render our bandit model far more complex than that in (Zhou et al. 2021). Unfortunately, the proposed policy in (Zhou et al.2021) assumed immediate feedback, thus cannot address this challenge. We also added experiments in paper revision to compare our policy UCB-FDF with UCB-List in Zhou et al. 2021 to corroborate our contributions. For the new experiments, please refer to Appendix II in our revised paper.
>
> > **Your Comment:** 2. In bandits with delayed feedback [Pike-Burke et al. 2018], their Theorem 2 illustrates that the regret bound has an additional term $\log(1/\Delta_j) E[\tau]$, which does not depend on $T$. However, in your Theorem 3, it has an additional term $\sqrt{4 E[\tau_1]\ln T}$, which has the dependence on $T$. Why is that? Can it be improved? Could you derive the lower bound and close the gap on the dependence of the delay period?
>
> **Our Response:** The difference in the bounds is due to a key difference in the delay assumptions: Pike-Burke et al. 2018 assumed a known expected delay, so that their proposed policy can leverage the knowledge of expected delay to make decisions, i.e., chioce of $n_m$ (see Section 4.1 in [Pike-Burke et al. 2018]). In comparison, our proposed policy does *not* assume any knowledge/information of the delay distribution. The additional term $\sqrt{4 E[\tau_1]\ln T}$ in our Theorem 3 can thus be viewed as the price to pay for not having the knowledge of the delay distribution. We note, however, that the knowledge of the delay distribution is difficult to obtain in general. Thus, our assumption is more practical than [Pike-Burke et al. 2018].
>
> For the upper bound of the regret, one of the key quantities that affects the regret analysis is $D^*(t)$, i.e., the maximum accumulated delayed feedback up to time $t$, and the upper bound of $D^*(t)$ leads to the additional term $\sqrt{4\mathbb{E}[\tau_1]\ln T}$. However, we are not aware of any immediately applicable results that can further sharpen the bound for this quantity. An improved understanding of $D^*(t)$ will help to improve the regret upper bound and close the gap to the regret lower bound.
>
> > **Your Comment:** 3. In Assumption 1, it assumes that the delays of arm a follows an independent delay sequence $\lbrace\tau_{a,t}\rbrace$, where each element is a random variable satisfying $\tau_{a,t}\sim T_a$. Can the result be generalized to the setting where $\tau_{a,t}$ follow different distributions when $t$ varies?
>
> **Our Response:** In Lemma 4, the upper bound of $D^*(t)$ requires the assumption that the delay distributions do not vary over time. While this assumption can be relaxed, we will require extra knowledge of delay, such as  the maximum delay or finite expected delay to avoid the impact of heavy tailed delay, and this relaxation could enlarge the regret upper bound.
>
> > **Your Comment:** 4. Why only the term $g(b,1)$ shows up in the regret bound but not $g(b,t)$ for $t>1$?
>
> **Our Response:** The model assumes that the incentive impact function $g(b,t)$ increases monotonically with time $t$, thus $g(b,1)\geq g(b,t), \forall t\in T$ and the regret is upper bounded by $g(b,1)$.
>
> Please continue to see our next response below.

---

> > ### Author Response · Authors · 2021-11-19
> > **Author Response (Continued)**
> >
> > > **Your Comment:** 5. Regarding the numerical experiments, how do you choose the self-reinforcing preference function $f(x)$ and the incentive impact function $g(b,t)$? In addition, can you compare the regret between the setting with delay effect and that without the delay effect, and different delay distributions, through which to illustrate the influence of the delayed feedback?
> >
> > **Our Response:** Our theoretical analysis requires the conditions $f(x)=\Theta(x^\alpha)$ with $\alpha>1$ and $g(b,t)>1$. To align with the conditions and have a clear observation of the experiments, we pick the simple functions $f(x)=x^\alpha$ with $\alpha=2$ and $g(b,t)=b=1.5$. We added experiments and discussions regarding comparisons over different system parameters in the paper revision. For the new experiments, please refer to Appendix II in our revised paper.

---

> > > ### Comment · Reviewer_mJjk · 2021-11-22
> > > **Reply to the authors**
> > >
> > > Thank you for the detailed response. The explanations clarify my previous questions.

---

### Official Review · Reviewer_Rqug · 2021-11-04

**Correctness:** 4
**Technical Novelty And Significance:** 3
**Empirical Novelty And Significance:** 3
**Recommendation:** 6
**Confidence:** 4

**Main Review:**

Strengths of the paper are as follows.

+ The paper considers a practical scenario where recommender systems face. Prior works considers only one of the aspects: delayed rewards, incentivizing exploration. This paper combines them to study the joint effects.

+ The paper is well-written with a meaningful experimental section where they empirically compare the incentive costs with regret.

+ The algorithm is realistic in terms of practical implementation.

The main weakness of this paper is as follows

- My main complain on the paper is primarily around positioning. In particular, it is not entirely clear to me what the main challenge for the new model is that do not exist in either the delayed reward MAB or the incentivized exploration lines of work. A discussion around that and why adapting the algorithm for incentivized exploration to handle delayed reward would not work? In particular, delayed rewards for an UCB type algorithm should not pose a whole lot of difficulties; so I am wondering why a new algorithm design is needed. Compaing and contrasting the current algorithm to prior works will greatly help the reader.

**Summary Of The Paper:**

This paper combines three aspects of MAB: delayed reward, incentivized exploration and self-reinforcing user preference. They motivate this problem from the perspective of online recommender systems. For this model, they propose a new UCB based algorithm that achieves the optimal upper bounds. They also setup an online experiment based on Amazon review data and show how the regret evolves for both arm-independent delays and arm-dependent delays.

**Summary Of The Review:**

As above I am supportive of this paper and line of work. I would like to better understand the challenges which will help me appreciate the results better. Thus, I would like the authors to address that.

---

> ### Author Response · Authors · 2021-11-19
> **Author Response**
>
> We sincerely thank the reviewer's constructive comments and valuable insights, which help improve the quality of our work significantly. We have carefully revised our paper according to your comments and suggestions. Please see our revised submission, where we have highlighted all major changes in **Blue color**. Please also see our point-to-point responses as follows:
>
> > **Your Comment:** 1. My main complain on the paper is primarily around positioning. In particular, it is not entirely clear to me what the main challenge for the new model is that do not exist in either the delayed reward MAB or the incentivized exploration lines of work. A discussion around that and why adapting the algorithm for incentivized exploration to handle delayed reward would not work? In particular, delayed rewards for an UCB type algorithm should not pose a whole lot of difficulties; so I am wondering why a new algorithm design is needed. Compaing and contrasting the current algorithm to prior works will greatly help the reader.
>
> **Our Response:** Yes, we agree that both delayed feedback impact and incentivizing exploration have been investigated in the bandit framework separately in the literature. The main and new challenge in our model is the combined effect of *self-reinforcing user perference* effect and delayed feedback. Specifically, delayed feedback in the bandit model coupled with self-reinforcing user preferences lead to the following new challenge: if a bandit policy does not handle the delayed feedback appropriately and makes decisions with outdated information, then sub-optimal arms will have the higher chance to collect more rewards, which would in turn lead to the higher user preferences on sub-optimal arms. This self-reinforing preferences on sub-optimal arms will lead to further sub-sampling on the optimal arm, which forms a *vicious cycle* and could even result in a linear regret if a sub-optimal arm reaches *monopoly*. Thus, the impacts of delay is far more complex than those in conventional delayed bandit models without self-reinforcing user preferences. This is the reason that adapting the algorithms for incentivized exploration to handle delayed rewards would not work and a new algorithm is needed.
>
> So far, the complex coupling and combined effect of feedback delay, self-reinforcing user preferences, and incentives have not been considered in existing bandit literature. Existing bandit policies that deal with delayed feedback also cannot  guarantee sub-linear total incentive costs. In this paper, our proposed policy judiciously compensates the delayed feedback in detecting arm-dominance and makes unbiased estimation on arms to guarantee the correct dominance of empirical optimal arm in the face of delayed feedback, while ensuring *logarithmic* total incentive payment at the same time. We also added experiments in paper revision to compare our policy UCB-FDF with existing policy (UCB-List in Zhou et al. 2021) to corroborate our contributions. For the new experiments, please refer to Appendix II in our revised paper.

---

### Official Review · Reviewer_wZfK · 2021-11-08

**Correctness:** 3
**Technical Novelty And Significance:** 3
**Empirical Novelty And Significance:** Not applicable
**Recommendation:** 6
**Confidence:** 4

**Main Review:**

While I find the problem setting to be intriguing, I have a number of concerns about the writing, proof techniques, and regret upper bounds. I will outline these concerns below:

**Discussion of related work**:

It seems to me that there are a number of typos in the related work discussing Bandits with Delayed Feedback. In particular, I was confused by the claim that Joulani et al. (2013) proves a regret bound of $O(\sqrt{K T \log T + K \mathbb{E}[\tau]})$, where $\tau$ is the delay. Indeed, this upper bound is false when $\tau = T$, since the regret necessarily is linear in this regime. However, Theorem 6 from that paper does indeed show a proper scaling in this regime. I would encourage the authors to clarify this subtlety somehow.

I was also confused by the regret lower bound citation in delayed feedback from Vernade et al. (2017). Indeed, a scaling of $\Omega(K\log(T))$ follows from the standard bandit setting. Perhaps you should clarify this?

Also, I was confused by the line “... where only an upper bound on the tail of the delay distribution is needed, without requiring the expectation to be finite.” Perhaps you should clarify that only a _polynomial_ upper bound on the tail of the delay distribution is needed here.

**Regret modeling and payments**:

It is not clear to me how the payments made by the bandit policy affect the performance of the algorithm in your model. Indeed, the regret is measured against a policy which makes an infinite payment to the best arm. Does this mean that the algorithm is also allowed to make unbounded payments and “no cost” to performance? If, instead, we measure regret with respect to a policy which can only make some bounded number of payments, how does the regret change? Perhaps there is some meaningful notion of “payment regret” that measures the excess number of payments over the genie policy?

**Algorithm naming**:

(Minor comment) Although your algorithm is called UCB-..., it seems to not actually be a UCB-style policy, as the exploration and exploitation phases are distinct (unless I am misreading something?). It seems this algorithm is more of an Elimination-style algorithm? Perhaps you could consider changing the name to reflect this.

**Technical concerns**:

I am confused by the claimed regret scaling in Lemma 1 and Theorem 3. Indeed, both bounds seem to have a scaling term on the order of $\Delta^* \cdot \mathbb{E}[D^*(T)]$. However, consider an environment where $\Delta^* = 1/\sqrt{T}$ (e.g., the standard minimax regret lower bound environment), and suppose that all delays are deterministically $T$, so that $D^*(T)=T$ a.s. Then, it seems that your Lemma and Theorem would give a regret upper bound of $O(\sqrt{T})$ in this case. However, it is clear that regret must be linear, since the policy never receives any rewards. Am I missing something? I do not see how the claimed scaling can be true.

In trying to understand where this scaling term comes from, I began reading the proof of Lemma 1. It seems that there is an issue in the inequality of (7) on page 13. Indeed, this inequality does not type-match, since the left-hand side is a deterministic quantity (just the probability of some event), but the right-hand side is random, as it depends on $T_a(t)$, $D_a(t)$, and $c_a(t)$. Note that there is a similar issue in equation (10). It seems that this analysis introduces the scaling term that does not make sense to me (discussed in the previous paragraph). Thus, I suspect that fixing this bug will change the reported regret scaling.

**Empirical results**:

I suggest that the authors consider adding at least some simple baseline to their experimental results. For example, you may consider adding a UCB baseline, to demonstrate that policies that ignore the incentive structure should achieve poor regret scaling. Additionally, it would be nice to include error bars in the plots.

----

_Post-author response_: Please refer to my response to the authors. I think that most of my main concerns have now been addressed.


**Summary Of The Paper:**

This paper studies a stochastic multi-armed bandit setting with delayed feedback, where additionally, the bandit policy must incentivize an external agent to pull the desired arm, and the arm preferences of this external agent are self-reinforcing. They design a policy, UCB-FDF, for this setting, and prove expected regret (and incentive payment) upper bounds under various delay distribution assumptions. Finally, they evaluate their policy on bandit environments constructed using Amazon review data.


**Summary Of The Review:**

Although the problem setting is interesting, there are a number of concerns I have with the writing and technical results/proofs of the paper (see above for details). Since I do not think these concerns can be sufficiently addressed during the rebuttal, I do not think that this paper is ready for publication.

----

_Post-author response_: Please refer to my response to the authors. I have changed my opinion of the technical concerns, and the remaining ones seem minor. The authors have given a satisfactory response, and I have increased my score accordingly. I would still like to see the concerns mentioned in my responses addressed before publication, however.

---

> ### Author Response · Authors · 2021-11-19
> **Author Response**
>
> We sincerely thank the reviewer's constructive comments and valuable insights, which help improve the quality of our work significantly. We have carefully revised our paper according to your comments and suggestions. Please see our revised submission, where we have highlighted all major changes in **Blue color**. Please also see our point-to-point responses as follows:
>
> > **Your Comment** 1. [Discussion of related work]**:**  It seems to me that there are a number of typos in the related work discussing Bandits with Delayed Feedback. In particular, I was confused by the claim that Joulani et al. (2013) proves a regret bound of $O(\sqrt{KT\log⁡ T+KE[\tau]})$, where $\tau$ is the delay. Indeed, this upper bound is false when $\tau=T$, since the regret necessarily is linear in this regime. However, Theorem 6 from that paper does indeed show a proper scaling in this regime. I would encourage the authors to clarify this subtlety somehow.
> >
> > I was also confused by the regret lower bound citation in delayed feedback from Vernade et al. (2017). Indeed, a scaling of $\Omega(K\log⁡(T))$ follows from the standard bandit setting. Perhaps you should clarify this?
> >
> > Also, I was confused by the line “... where only an upper bound on the tail of the delay distribution is needed, without requiring the expectation to be finite.” Perhaps you should clarify that only a *polynomial* upper bound on the tail of the delay distribution is needed here.
>
> **Our Response:** Thanks for your comments. As you suggested, we would like to clarify the subtlety regarding the regret bounds you pointed out:
>
> * The confusion in Joulani et al. (2013) is due to a typo (our apologies!). The worst case regret bound of [Joulani et al. 2013] should be written as $O(\sqrt{KT\log T}+K\mathbb{E}[\tau])$ (the square root was too long in our initial submission). This worst-case regret is directly obtained from their Theorem 6 by plugging in the worst-case delay (i.e., $\mathbb{E}[\tau]=T$) as you suggested. Thus, Joulani's result also shows that, with worst-case delay being $\mathbb{E}[\tau]=T$, the expected regret grows linearly. Please note that this is reasonable since, with expected delay being $T$ (expected delay grows without bound as the time horizon $T$ increases), no policy can achieve sublinear regrets due to the significant information loss caused by such a large delay. We have fixed the typo of the regret bound (please see Page 3, Section 2.1) in our revised submission). On the other hand, we would also like to point out that our paper assumes bounded expected delay ($\mathbb{E}[\tau]$ does not grow with $T$, i.e., $\mathbb{E}[\tau] \ne T$). Thus, the probability of $\tau=T$ is *increasingly rare* as $T$ gets large. Thus, the expected regret of our policy still grows *sublinearly* even though the worst-case regret is linear.
> * Yes, the work in [Vernade et al. (2017)] showed that, under their model, a regret scaling of $\Omega(K\log T)$ is achievable. Note that this result is interesting because delayed bandit problems are harder in general due to the loss of information resulted from delays. Thus, it is not immediately clear whether the $\Omega(K\log⁡(T))$ lower bound of the standard bandit setting continues to hold in the delayed settings. Yet, the work in [Vernade et al. (2017)] showed that the answer is yes.
> * The line "*... where only an upper bound on the tail of the delay distribution is needed, without requiring the expectation to be finite.*" was describing the setting in [Manegueu et al. 2020], where we meant their delay distributions are possibly *heavy-tailed* (including polynomial as you pointed out). We have clarified this in our revised submission (see Page 3, Section 2.1)).
>
> Please continue to see our next response below.

---

> > ### Author Response · Authors · 2021-11-19
> > **Author Response (Continued)**
> >
> > > **Your Comment** 2. [Regret modeling and payments]**:** It is not clear to me how the payments made by the bandit policy affect the performance of the algorithm in your model. Indeed, the regret is measured against a policy which makes an infinite payment to the best arm. Does this mean that the algorithm is also allowed to make unbounded payments and “no cost” to performance? If, instead, we measure regret with respect to a policy which can only make some bounded number of payments, how does the regret change? Perhaps there is some meaningful notion of “payment regret” that measures the excess number of payments over the genie policy?
> >
> > **Our Response:** Thanks for your interesting comments. First, we clarify that our policy is not allowed to make unbounded payment. Coupled with the fact that our policy doesn't have the knowledge of the optimal arm in hindsight, online learning regret is inevitable under our proposed policy.
> >
> > Regarding your suggested "payment regret", we note that introducing the notion of "payment regret" is somewhat unnecessary in our settings, although doing so is seemingly intuitive. This is due to the following reasons:
> >
> > * It is clear that any policy with bounded payment cannot outperform the genie policy that has an infinite amount of payment to incentivize the optimal arm. Thus, the "payment regret" of our proposed policy (compared to an optimal bounded-payment policy) must be *smaller* than the "genie regret" of our proposed policy (compared to the genie policy).
> > * In this work, we showed that the "genie regret" of our proposed policy is $O(\log T)$, which is already lower-bound matching (since the lower bound of delayed bandit is also $O(\log T)$, see our response to your previous comment). Combining this and the fact that "payment regret" is smaller than "genie regret" (see the previous bullet), it immediately follows that the "payment regret" of our proposed policy is also upper bounded by $O(\log T)$. In other words, the "payment regret" must also be order-optimal.
> >
> > Although studying "payment regret" is unnecessary, we still thank you for your insightful comments. In this revision, we have added the above discussions (see Page 5, Section 3.3).
> >
> > > **Your Comment** 3. [Algorithm naming]**:** (Minor comment) Although your algorithm is called UCB-..., it seems to not actually be a UCB-style policy, as the exploration and exploitation phases are distinct (unless I am misreading something?). It seems this algorithm is more of an Elimination-style algorithm? Perhaps you could consider changing the name to reflect this.
> >
> > **Our Response:** Thanks for your suggestion. We name our policy by "UCB-Filtering-with-Delayed-Feedback" because the arm filtering mechanism in the exploration phase of our policy is based on comparing the arms' upper confidence bounds, hence the name. Note that the same naming convention has also been used in the literature. For example, the "KL-UCB" policy in [Kaufmann and Kalyanakrishnan, COLT'13] is also named after the UCB-based arm removal. Due to this reason, we will still keep the policy's name in this revision.
> >
> > Please continue to see our next response below.

---

> > > ### Author Response · Authors · 2021-11-19
> > > **Author Response (Continued)**
> > >
> > > > **Your Comment** 4. [Technical concerns]**:** I am confused by the claimed regret scaling in Lemma 1 and Theorem 3. Indeed, both bounds seem to have a scaling term on the order of $\Delta^* \cdot\mathbb{E}[D^*(T)]$. However, consider an environment where $\Delta^*=1/T$ (e.g., the standard minimax regret lower bound environment), and suppose that all delays are deterministically $T$, so that $D^*(T)=T$ a.s. Then, it seems that your Lemma and Theorem would give a regret upper bound of $O(T)$ in this case. However, it is clear that regret must be linear, since the policy never receives any rewards. Am I missing something? I do not see how the claimed scaling can be true.
> > > >
> > > > In trying to understand where this scaling term comes from, I began reading the proof of Lemma 1. It seems that there is an issue in the inequality of (7) on page 13. Indeed, this inequality does not type-match, since the left-hand side is a deterministic quantity (just the probability of some event), but the right-hand side is random, as it depends on $T_a(t)$, $D_a(t)$, and $c_a(t)$. Note that there is a similar issue in equation (10). It seems that this analysis introduces the scaling term that does not make sense to me (discussed in the previous paragraph). Thus, I suspect that fixing this bug will change the reported regret scaling.
> > >
> > > **Our Response:** Thanks for your question. It appears that there are some misunderstandings of the bandit setting in our paper. We would like to further clarify as follows:
> > >
> > > * In this paper, we do not consider minimax regret. Instead, we focus on developing incentivized bandit policies that are lower-bound matching in terms of conventional pseudo regret (compared to the genie policy). Moreover, the mean values of arm rewards are constants and *not* time-varying. That is, the $\mu_a$-values of each arm $a$ is fixed and does not change as the time horizon $T$ gets large. As a result, $\Delta$ is also a constant and does not shrink as $T$ gets large.
> > > * In this paper, we assume finite expected delay. As a result, although the random delay $\tau$ could still be as large as $T$, $\tau=T$ is increasingly rare as $T$ gets large. Thus, "$D^*(T) = T$ a.s." cannot happen in our bandit setting.
> > >
> > > On the other hand, in the case of constant delay $\tau=T$, any policy (including ours) will have a linear regret due to the significant loss of feedback information (please also see our response to your Comment 1).
> > >
> > > Regarding your comments on "type matching" in Eq.(7) on Page 13, we note that there is no error in this equation and there seems to be some confusion in notation. The left-hand-side of Eq.(7) is the probability of some event (as you correctly pointed out). The right-hand-side of Eq.(7) is also a deterministic value because the $T_a(t)$, $D_a(t)$, and $c_a(t)$ represent the *realizations* of the corresponding random variables in this case for lighter notation (i.e., $D_a(t) =$ some value). Although this is indeed a slight abuse of notation, it is a convention in the literature (see, e.g., the bandit survey by [Bubeck et al. 2012]). Also, the inequality in Eq.(7) follows directly from the Hoeffding Inequality.
> > >
> > > > **Your Comment** 5. [Empirical results]**:** I suggest that the authors consider adding at least some simple baseline to their experimental results. For example, you may consider adding a UCB baseline, to demonstrate that policies that ignore the incentive structure should achieve poor regret scaling. Additionally, it would be nice to include error bars in the plots.
> > >
> > > **Our Response:** Thanks for the suggestions! We added experiments and discussions regarding baseline comparison in the paper revision, where the baseline is the policy UCB-List in Zhou et al. 2021. We also added experiments regarding delay distribution comparisons and system parameter comparisons in the paper revision. Considering the information density, instead of error bars in the plots, we provided jittered plots of all the random cases in Figure 1. For the new experiments, please refer to Appendix II in our revised paper.

---

> > > > ### Comment · Reviewer_wZfK · 2021-11-28
> > > > **Reply to authors (continued, 2)**
> > > >
> > > > 4) Regarding technical concerns. Thank you for your response, and upon further reflection, I am much less concerned with the points I had initially raised. However, I am slightly confused by a couple of your responses (though my confusion is on somewhat minor points).
> > > >     - Finite expected delay setting. I do not see why your Assumption 1 precludes the setting where $D_a(t) = T$ deterministically, since $T$ is arbitrarily large, but not infinite, so this seems to fit into your problem setting. Additionally, you say that $\Delta_a$ does not depend on $T$ in your setting -- however, I don't think any of your assumptions in your paper preclude this scenario. **Neither of these points**, however, seem to be problematic, and it appears that the points I raised in my comment don't quite make sense. I suppose I was thinking that the number of suboptimal arm pulls when $D^* = T$ should be linear (which is indeed the case in your analysis). But in the scenario I described, it is clear that the regret should be a constant, simply by the standard regret decomposition/Wald identity (e.g., Lemma 4.5 in [Lattimore and Szepesvári, 2020]). The scaling your results have w.r.t. the expected delay makes a lot of sense to me, and I apologize for this confusion!
> > > >     - Type matching. While I agree that arguments involving a random number of samples are standard within the bandits literature, I disagree that such notation is standard. Indeed, Hoeffding's inequality is true for a _fixed_ number of samples, whereas you seek a more subtle concentration for a _random_ number of samples, where this random number is _not_ independent of the samples (since the number of times an arm is played depends on the previously observed samples). I think what the authors intend to write on the LHS is the probability _conditioned on $T_a(t) = s$_. However, it is not clear to be that the rewards remain independent, conditioned on this event, since (as mentioned earlier) the number of times an arm is pulled depends on the previously observed samples. **This being said**, I think that this issue is largely technical, and can be reasonably easily overcome. In the worst case, you should be able to apply the arguments from Exercise 7.1 (c) in the Bandit Algorithms book [Lattimore and Szepesvári, 2020] to prove the statement you need, perhaps with a minor modification to the confidence sequence as used in this exercise. I would suggest that the authors make these changes before camera-ready (which should be reasonably easy to do I think), since I still think that the expression does not type-match.
> > > >
> > > > 5) Regarding experimental evaluation. Thanks for adding these additional experiments and comparisons! You may consider including the comparison with the UCB-List somewhere in the main body (maybe even in the same plot?).
> > > >
> > > > ----
> > > >
> > > > _In summary_: After reading the authors' responses to my and the other reviewers' concerns, I think that my initial assessment of the paper was overly harsh, as my main concerns seem not to be of major concerns. While there are some remaining concerns (highlighted above) that I would like to see addressed, I believe that these can be reasonably done before the camera ready. I especially encourage the authors to address the concerned mentioned in my second bullet of (4) Technical concerns, regarding the use of Hoeffding's inequality. Thus, I will increase my score to be consistent with the other reviewers, as a Weak Accept.

---

> > > ### Comment · Reviewer_wZfK · 2021-11-28
> > > **Reply to authors (continued)**
> > >
> > > 2) Regarding the expected payment. This makes a lot of sense, thanks for the clarification. I agree that $\mathcal{O}(\log(T))$ expected payment makes sense as scaling, and perhaps it would be useful to add this comment somewhere in your paper (unless I have missed it in the submission already?). However, it is not 100% clear to me that there should also be an $\Omega(\log(T))$ payment regret lower bound, since perhaps there is a policy that does not need to incentivize _any_ arm during some early rounds (just as your policy stops incentivizing in the self-sustaining phase). If there is an argument for why such a policy would achieve linear (or suboptimal) regret scaling, this might be useful to include as a discussion point also.
> > >
> > > 3) Regarding algorithm naming. (Again, just a very minor comment) KL-UCB doesn't come from the paper you refer to (as I understand, this policy was simultaneously proposed by  Garivier and Cappe [2011] and Maillard et al. [2011]). Did you mean to refer too the KL-Racing algorithm from [Kaufmann and Kalyanakrishnan, COLT'13]? Their other algorithm, KL-LUCB, does seem to be selecting arms based on optimistic arm estimates. In my view, UCB is an alg which optimistically selects arms at each round, and elimination algorithms maintain a set of arms, and iteratively remove based on data-dependent (possibly optimistic mean estimate) criteria. In this sense, I find the UCB naming a bit misleading. Anyways, just a suggestion, and not a big deal.

---

> > ### Comment · Reviewer_wZfK · 2021-11-28
> > **Reply to authors**
> >
> > I would like to thank the authors for their detailed and thoughtful responses to my questions (and apologize for the delay in my reply). I will address the responses in the same order as before.
> >
> > 1) Regarding related work. I appreciate the edits to this section, and it clears up most of the points I was confused on. I'm still a bit confused on the regret lower bound cited in [Vernade et al. (2017)] though, since, as you mention, the delayed feedback is a strictly harder model than standard bandits, so it is still not clear to me why the standard Lai & Robbins lower bound does not carry over immediately to their setting (Theorem 4 in their paper -- though their lower bound result Theorem 3 makes sense). This is just a confusion on their paper at this point though, so it's not really a big deal, and it seems totally fine to mention. Perhaps you could add that this work obtains strictly larger regret lower bound in some feedback settings (e.g., Theorem 3 of their work)? Just a suggestion -- this is up to you.

---

> ### Author Response · Authors · 2021-11-30
> **Author Responses to the Second-Round Comments from Reviewer wZfK**
>
> We appreciate the reviewer's constructive comments and suggestions. For the reviewer's follow up questions, please see our responses below (for better readability, we respond to all your second-round comments in one place):
>
> 1. **Confusions of Lower bounds in [Vernade et al. (2017)].** In our understanding, also as indicated in [Section 4.2, Vernade et al. (2017)], the proposed lower bound in the "uncensored delays" setting (Theorem 4) does not fundamentally differ from the Lai & Robbins lower bound (i.e., $\Omega(\log T)$), which is due to the consideration of "uncensored delay" effects. However, as you correctly pointed out, the regret lower bound in the "censored delay" setting (Theorem 3) may be higher than the Lai & Robbins lower bound (due to the seemingly harder assumption that a reward-conversion can either be observed only within $m$ time steps after the action occurred or lost). This implies Lai & Robbins is at least a loose lower bound. Thus, the "surprise" lies in what they showed that Lai & Robbins' lower bound is actually tight. In the final version, we will again clarify the lower bounds in [Vernade et al. (2017)].
>
> 2. **Payment Regret Lower Bound.** Thanks for your questions. But it seems there remains some confusion here. In our previous rebuttal, we didn't mean a policy with bounded payment "would achieve linear (or suboptimal) regret scaling." What we actually meant in our previous rebuttal is the following:
>    * Let $\Gamma_g$ and $\Gamma_{bp}$ denote the optimal rewards achieved by the genie and the optimal bounded-payment policies, respectively. Also, let $\Gamma_p$ denote the achievable reward by our proposed policy.
>    * Since the optimal bounded-payment policy cannot outperform the genie policy (with infinite payment and knowledge of the optimal arm in hindsight), we have $\Gamma_{bp} \leq \Gamma_g$. It thus follows that, for the bounded-payment regret $R_{bp}$ and the genie regret $R_g$, we have $R_{bp} \triangleq \Gamma_{bp} - \Gamma_p \leq \Gamma_g - \Gamma_p \triangleq R_g \stackrel{(a)}{=} \mathcal{O}(\log T)$, where the equality $(a)$ is proved in our paper. Thus, $R_{bp}$ is also order-optimal and hence the notion of bounded-payment regret may not be necessary. Hope this simple derivation can clarify the confusions.
>
> 3. **Algorithm Naming.** Thanks for sharing your views! We do see your point that our algorithm naming could be odd to some people. We will consider renaming our algorithms in our final camera-ready version to avoid such confusion.
>
> 4. **Technical Concerns.** Thanks for your further comments. Please see our responses below:
>    * *Finite Expected Delay Setting.* We are glad that we are finally on the same page and you agree with our results! Yes, our assumption on finite expected delay is indeed important. We will definitely highlight this important finite expected delay assumption in our final version.
>    * *Type Matching.* Thank you for the suggestion and the technical reference! We will try to clarify Eq. (7)   in our final version following the approach in [Lattimore and Szepesvári, 2020] as you suggested.
>
> 5. **Experimental Evaluation.** Thanks for the suggestion! We will move the comparison results with UCB-List to the main body of the paper in our final version.

---

### Decision · Program_Chairs · 2022-01-20

**Decision:**

Accept (Poster)

**Comment:**

This paper tackles a bandit problem that incorporates three challenges motivated by common issues encountered in online recommender systems: delayed reward, incentivized exploration, and self-reinforcing user preference. The authors propose an approach called UCB-Filtering-with-Delayed-Feedback (UCB-FDF) for this problem and provide a theoretical analysis showing that UCB-FDF achieves the optimal regret bounds. Their analysis also implies that logarithmic regret and incentive cost growth rates are achievable under this setting. These theoretical results are supported by empirical experiments, e.g. using Amazon review data. The main concern with this paper is that the considered challenges have all been tackled already in different bandit settings, so the novelty here is that they are being tackled altogether. It would be more convincing if experiments included baselines from these existing settings to motivate the need for a new strategy rather than simply relying on methods that have been proposed previously to address each of these problems independently; the experiments currently contain only a baseline for bandits with self-reinforcing user preference, which has been added during the rebuttal phase.